# Interventions to Promote the Utilization of Physical Health Care for People with Severe Mental Illness: A Scoping Review

**DOI:** 10.3390/ijerph20010126

**Published:** 2022-12-22

**Authors:** Michael Strunz, Naomi Pua’nani Jiménez, Lisa Gregorius, Walter Hewer, Johannes Pollmanns, Kerstin Viehmann, Frank Jacobi

**Affiliations:** 1Psychologische Hochschule Berlin, 10179 Berlin, Germany; 2Institute for Health Services Research and Health Economics, German Diabetes Center, Leibniz Center for Diabetes Research, Heinrich Heine University Duesseldorf, 40225 Duesseldorf, Germany; 3Klinikum Christophsbad, 73035 Göppingen, Germany; 4Kassenärztliche Vereinigung Nordrhein, 40182 Duesseldorf, Germany; 5Institute for Health Services Research and Health Economics, Centre for Health and Society, Medical Faculty, Heinrich Heine University Duesseldorf, 40225 Duesseldorf, Germany

**Keywords:** severe mental illness (SMI), physical health promotion, utilization of health care, access to health care, collaborative care, integrated care

## Abstract

Background: The main contributor to excess mortality in severe mental illness (SMI) is poor physical health. Causes include unfavorable health behaviors among people with SMI, stigmatization phenomena, as well as limited access to and utilization of physical health care. Patient centered interventions to promote the utilization of and access to existing physical health care facilities may be a pragmatic and cost-effective approach to improve health equity in this vulnerable and often neglected patient population. Objective/Methods: In this study, we systematically reviewed the international literature on such studies (sources: literature databases, trial-registries, grey literature). Empirical studies (quantitative, qualitative, and mixed methods) of interventions to improve the utilization of and access to medical health care for people with a SMI, were included. Results: We identified 38 studies, described in 51 study publications, and summarized them in terms of type, theoretical rationale, outcome measures, and study author’s interpretation of the intervention success. Conclusions: Useful interventions to promote the utilization of physical health care for people with a SMI exist, but still appear to be rare, or at least not supplemented by evaluation studies. The present review provides a map of the evidence and may serve as a starting point for further quantitative effectiveness evaluations of this promising type of behavioral intervention.

## 1. Background and Rationale

People with a severe mental illness (SMI) have an increased risk of mortality, compared to the general population. Two-thirds of these premature deaths are due to somatic causes, mostly as a consequence of physical disease [1]. The causes for this are diverse and routed in the healthcare system, healthcare providers, and factors related to people with a SMI. These causes include the fragmentation and insufficient integration of health care systems, adverse effects of drug interaction, diagnostic overshadowing (a process by which physical symptoms are misattributed to mental illness), stigmatization phenomena (including self-stigmatization), and unfavorable, or ineffective health behaviors and utilization of physical health-care services [2]. Concepts that address these issues, in order to promote the physical health of people with a SMI, are known as integrated or collaborative care approaches [2,3]. Their aim is to reduce the fragmentation and to promote the coordination of health care. As people with a SMI often suffer from complex mental and physical health issues, this group particularly benefits from integrated and collaborative health care [4]. Such concepts often integrate primary care providers in local mental health services and vice versa, with the aim to better coordinate health care for people with complex health needs [5,6,7]. However, integrative health care services need an appropriate infrastructure, which is rarely readily available [8].

Interventions which mainly focus on the patient side to directly improve the physical health of people with a SMI have often been explored, in terms of lifestyle interventions [9,10,11,12]. These interventions focus on the promotion of healthy lifestyles and wellness, including structured approaches to change physical activity, weight, healthy diet, and healthy habits (e.g., quit or reduce smoking). Going beyond, some interventions indirectly improve the physical health of people with a SMI by promoting the utilization of health care services without extensive changes in the health-care system itself. Collaborative health care appears to be a promising approach for establishing equitable opportunities in physical health for people suffering from a SMI. It includes the referral to already existing prevention- and health-promotion-programs that are available for the general population [3].

To develop and refine our research objectives, we identified 37 reviews [3,5,6,7,9,10,11,12,13,14,15,16,17,18,19,20,21,22,23,24,25,26,27,28,29,30,31,32,33,34,35,36,37,38,39,40,41] focusing on the physical health of people with a SMI, using a forward and backward reference search. None of them systematically investigated the promotion for service use, as an important sub-concept of integrated care which therefore is subject of the present review. For this purpose, the format of a scoping review is particularly useful, since we assumed a lot of heterogeneity across the interventions under investigation. Furthermore, we want to provide an overview about the current research state and research gaps, as comprehensively as possible, without the limitation of a formal assessment of the methodological study quality.

This review will scope the literature on the interventions promoting the utilization of physical health-care services for people with a SMI. To identify the relevant studies, we follow the PCC (population/concept/context) framework [42] to determine the target population, concept, and context. The details are given in the methods section.

Objectives: Based on these guiding principles, this scoping review seeks to answer the following research questions:(1)How many and what types of patient-centered interventions have been examined or designed to promote the utilization of physical health care systems for people with a SMI?(2)Which studies report positive outcomes on the utilization behavior, as well as on physical health measures, and what are promising concepts for the broader implementation into health care systems?(3)What are current research gaps regarding this topic?

## 2. Materials and Methods

Protocol and registration: The scoping review strictly follows the guidelines of von Elm et al. [43] and the extended PRISMA criteria for scoping reviews [44]. The literature search, search strategy, as well as the inclusion and exclusion criteria for the studies were undergoing an iterative process and were documented continuously. We registered a final protocol, prospectively (Open Science Framework on 22 May 2022 [45]).

Eligibility criteria: We applied the following inclusion and exclusion criteria for the PCC components, in order to identify the relevant studies. (1) Population: The target group for the study were adults (age 18+) with a severe mental illness, defined as psychotic disorder, bipolar disorder, and severe depression; further we included borderline personality disorder, due to its high rate of physical comorbidity [46,47]. There were no exclusion criteria, based on other disorders (e.g., substance use disorder, posttraumatic stress disorder, obsessive compulsive disorder), if the target population was defined as severely mentally ill and included at least one of the above-mentioned diagnoses. As we focus on people who have general access to their specific health care system, interventions for homeless persons were excluded. In addition to our protocol, we excluded studies examining different populations, but without distinct outcome reports for the SMI population. (2) Concept: Studies that were included described interventions with the explicit aim of promoting the use of and access to pre-existing physical health care and preventive services (e.g., pre-existing health promotion programs). Acknowledging the nature of complex health interventions, additional interventions could be part of a study, as long as the promotion of health service use was included. Interventions aimed directly at promoting physical health (e.g., lifestyle interventions) without the utilization promotion, were excluded. Studies focusing solely on improving mental health were also excluded.

The concept under review implies a person-centered approach. Therefore, we also excluded studies that focused only on the physical health of people with mental illness at the provider level (e.g., training of health care professionals). If complex interventions additionally included elements at the provider level, in addition to the person-centered utilization promotion, this was not a reason for exclusion. (3) Context: Only studies published between January 2000 and April 2022, were included, in order to identify studies which can be applied on current health care systems. As the aim of this scoping review was the identification of studies without significant modifications of the health care settings itself, or without the creation of a new care structure, we excluded the intervention settings with co-located mental and physical health care services (e.g., behavioral health home settings [18]). Thus, we focused on the low threshold promotion interventions without implementing new medical care resources, in light of the limited infrastructure or restricted health budgets (especially in rural areas; [48]). In addition to our protocol, studies which only took place in an inpatient setting had to report outcomes after the end of hospitalization, to be included in the review.

We included all studies reporting results on changes of the utilization of the physical healthcare system or on physical health status, or studies that were planning it (protocols and trial registries). Quantitative, qualitative, and mixed method designs were eligible. Studies without interventions (e.g., observational studies or guidelines) and secondary research studies (e.g., overviews or reviews) were excluded.

Information sources and search: (1) We conducted our search using the databases Web of Science Core Collection and PubMED. The search was limited to English and German articles published between January 2000 and April 2022. The search used a combination of the search terms for each database using Boolean operators (e.g., ‘and’, ‘or’). Keywords for population, concept, and context were extracted beforehand from the existing reviews of integrated care concepts for people with a SMI. We used several particularly interesting studies identified beforehand through the existing reviews, to test the appropriateness of our search strategy. All reviewers came to an agreement for the final search strategy. The detailed information on the search strategy for each source is reported in Appendix A. (2) Furthermore, unpublished literature was searched using the platforms https://easy.dans.knaw.nl/ (accessed on 13 June 2022) and http://catalog.crl.edu/search~S4 (accessed on 05 July 2022). The former is a platform for research data storage which is mainly used by Dutch researchers. We consider the Netherlands as particularly relevant for the evaluation and implementation of new concepts in public health, and therefore chose this platform as potential source for unpublished literature. The latter is a platform which lists dissertations, another potential source for unpublished projects. (3) We also conducted a search on the trial register platforms https://trialsearch.who.int (accessed on 27 June 2022) and https://clinicaltrials.gov (accessed on 20 June 2022), in order to identify not yet published and unpublished studies.

Selection of sources of evidence: All search results were transferred to the reference management tool Zotero, double entries were removed. To increase consistency among reviewers, in a “training phase”, the first 100 publications were screened by M.S. and N.P.J., and their disagreements were documented and subsequently discussed among the six reviewers F.J., J.P., K.V., L.G., M.S., and N.P.J., until we reached consensus. We then conducted the review process in two stages and every entry was screened by two reviewers, independently in both stages: First, F.J., J.P., K.V., L.G., M.S., and N.P.J. selected studies, based on title, abstract, and on the defined inclusion- and exclusion-criteria. In case of disagreement, the studies were included in the full-text screening stage to maximize the sensitivity of the screening process. The full-text of relevant studies was checked by the same reviewers for the final inclusion. To reach a consensus in cases of non-conformity in the second stage, a further reviewer (W.H.) was consulted.

Data charting: The data was extracted integrating the recommendations of Arksey and O’Malley (2005) [49]. The data-charting form was jointly developed by F.J. and M.S. M.S. charted the data, and this extraction was double-checked by N.P.J., K.V., and J.P. Within this process, we added the extraction item “theoretical rationale or model” because the evidence-based interventions in health should be designed using a well-established psychological, behavioral framework that is used to guide the intervention [6,50,51], and thus we decided to include this aspect in our scoping review.

Data items: Standardized Excel-spreadsheets with sections author, year, country, study design, study aim, context or setting of intervention, sample characteristics, interventions, theoretical rationale or model, outcome measures, main results, and limitations were implemented.

Synthesis of the results: We conducted a narrative synthesis of the general study characteristics. We synthesized the different types of reported study interventions, as well as their underlying theory, model, or framework if found. All intervention types, theoretical rationales, and reported outcomes of the included studies, are presented along with the study identification numbers (from Appendix A) in order to guide the reader from the aggregated tables to the individual studies.

## 3. Results

### 3.1. Study Selection and Exclusion Process: Number of Identified Studies

From n = 1773 records identified in the database search, n = 51 publications [52,53,54,55,56,57,58,59,60,61,62,63,64,65,66,67,68,69,70,71,72,73,74,75,76,77,78,79,80,81,82,83,84,85,86,87,88,89,90,91,92,93,94,95,96,97,98,99,100,101,102] were included in the synthesis (Figure 1). During the selection process we identified three more publications of studies that at that point had been included in our records only as study protocols. Finally, the identified 51 publications referred to 38 different studies (i.e., some studies were represented in more than one publication).

### 3.2. Study Characteristics

The general study characteristics including the description of the author, year, country, study design, aim, intervention setting, and target group (along with the theoretical rationales, outcome measures and results, see below) are described in the comprehensive Appendix A.

Included were four study protocols aiming to describe the intervention and their investigation, without having presented results yet. We included studies from 10 different countries. We identified 24 studies that were conducted in the U.S., nine studies in Europe, three in Australia, and one study in a non-OECD Country (Thailand). In 35 studies (including three with a study protocol only), quantitative research designs were employed, two studies used qualitative research designs, and three studies (including one with a study protocol only) used mixed method designs. The study aims were mostly an investigation of the intervention outcomes (25), or an investigation of the intervention feasibility (8). Settings of intervention were heterogenous and included primary care settings (4), local mental health services (20), veterans affairs mental health centers (2), or were conducted at the patients’ home or a setting of their choice (5). We found homogenous SMI-diagnoses (e.g., only individuals suffering from bipolar disorder) in nine study samples. Twenty study samples included individuals with two or more different SMI diagnoses. SMI diagnoses were not specified in 11 study sample descriptions. Only one study included an explicitly borderline personality disorder diagnosis, and four studies labeled obsessive-compulsive disorder (OCD) and post-traumatic stress disorder (PTSD) as further SMI diagnoses. In 18 studies, individuals with a SMI were only included if they already had developed physical health risk factors. Sample sizes of the included investigations (intervention group only) ranged from 9 to 4788 individuals. 11 studies involved investigations with 9–50 individuals, 11 samples were moderately sized with 51 to 100 individuals, 12 samples involved sizes with 101 to 200 individuals, and four samples involved larger sample sizes of 205, 252, 857 (4788) and 988 individuals.

### 3.3. Interventions, Background, and Outcomes

Each identified intervention type is presented with the respective studies in Table 1. Studies usually investigated complex interventions, which means that the promotion of health care utilization was not the only intervention concept. As the implementation of intervention types differs in each study, the details of a particular study and its different intervention elements can be found in Appendix A. For example, the intervention type ‘training on the better utilization of health care’ in study 1, was part of nine group training sessions delivered by health care professionals and peer specialists, targeting mainly the ability to communicate with health care providers. In study 23, this type of intervention was implemented through weekly and individual coaching sessions delivered by health professionals. These helped participants to improve their ability to engage with healthcare providers to address their health comorbidities.

In addition to the content of interventions, we were interested in how interventions were delivered (e.g., in what format). Interventions were mostly delivered through an individual setting (37), 17 of them with an additional group setting. One intervention was only delivered in a group setting (study 4). We identified an explicit phone maintenance in two interventions, and one intervention was app-based. Peer specialists were implemented in six studies, four of them were peer-led without additional health care professionals. Most common elements of interventions included care-management (27), self-management support (23), physical health education (16), health screening and monitoring (16), lifestyle changes (12), training on the better utilization of health care (13), mental health education (10), care-plan development (9), and staff training (9). The duration of the interventions ranged from two to 24 months, most included six (8) or 12 (13) months of duration. In one study, the overall duration of the interventions remained unclear. The frequency of the conducted interventions ranged from daily (1) to every three months (5), with most of interventions being conducted weekly (6) to monthly (12). The frequency of interventions remained unclear in 11 studies.

The types of outcome measures are listed in Table 2. Most studies included self-reported outcomes (35), nine of them with self-reported data only. Most common self-reports were related to the mental and physical health related quality of life, health care service use, adherence and satisfaction with the intervention, health behavior and health knowledge, and behavioral, social, and self-management skills. We identified 18 studies that included physiological measures, such as cardiovascular health parameters. Administrative data, such as health insurance registries were used in 10 studies. In addition to our analysis of the reported outcome measures, we were interested in the potential follow-up measurements of outcomes after the end of the respective intervention. Only three studies (study 5, study 23, and study 35) included this data assessment design.

The type of reported main findings on the interventions’ success are listed in Table 3 (note: this table refers only to the 33 of the 38 studies that have published their results). In terms of the reported outcomes and their interpretation by the authors of the respective studies, improvements in physical health-behavior related outcomes (e.g., self-management or patient-activation) were reported for 14 studies. Improvements in physical health outcomes (self-reported and not self-reported, e.g., physical health related quality of life or blood pressure), were reported in 13 studies. Although the main target of the studied interventions, only 10 reported improvements in the health service utilization itself (based on the self-reported as well as administrative data). To include mention of the effects beyond the service use, improvements in mental health related outcomes (e.g., self-esteem, psychiatric symptom burden, mental health related quality of life) were reported in 10 studies, and four studies reported a reduced emergency department use. In six out of 33 studies we did not identify positive results for any of the above-mentioned health related outcomes at all.

The description of the specific underlying theoretical rationales by study (that were reported in about 75% of the studies) is provided in Table 4. The theoretical background of the interventions was based on the formerly evaluated interventions in 17 cases. We identified adaptions of complex care models (most commonly the chronic care model (5) and the assertive community treatment model (4)) in 13 studies, seven studies were based on one or more different behavioral, motivational, social, and cognitive theories, and five studies did not report any model or theoretical rationale. An additional theoretical framework for the implementation of the intervention was reported in six studies.

## 4. Discussion

### 4.1. Available Studies on the Interventions to Promote the Utilization of Physical Health Care in SMI

The need for health prevention and promotion of health care has been highlighted by numerous studies investigating poor physical health outcomes in individuals with severe mental disorders, e.g., [1,2,50,51,103,104]. Since the lack of interventions at the provider level has already been extensively documented (e.g., regarding the prevention of diagnostic overshadowing [105]) we wanted to examine the interventions at the patient level. In the present scoping review, we therefore focused on the systematic efforts to promote the SMI-patients’ utilization of physical health care services and summarized a variety of interventions, along with their rationales and outcome measures. We could identify 38 studies (mostly from the U.S., only one from a non-OECD country) that investigated how the utilization of existing care (i.e., beyond special programs of integrated care designed for individuals with a SMI) can be promoted. That exceeds the findings of a scoping review provided by Richardson et al. (2020) [6], who had identified and investigated 25 studies on the broader concept of physical and mental health care integration. Nonetheless, it still highlights a research gap in this field against the background of the magnitude of excess mortality and health inequity in SMI. This gap is also visible in the very unequal regional distribution of studies, even though the prevalence of SMI is relatively homogenous all over the world.

### 4.2. Implementation of the Access and Use Promotion: Promising Concepts and Major Gaps

Most of the identified studies involved complex interventions which were not always easy to disentangle. The majority of interventions included individual support (in particular care-management and self-management strategies), sometimes supplemented by group formats. They seem to be valuable complementary elements, in addition to aspects of coordinated physical and mental healthcare, for people with SMI. In order to bridge the physical health-related gaps for this population, more complex and multifactorial interventions seem to be promising, including further elements, such as coordination with community resources, continuous familiarity with service users over time, and the delivery of person-centered care [6,41].

Supplementation and maintenance by phone or digital tools were surprisingly rare; this suggests much room for improvement regarding the cost-effectiveness of interventions (given that the digital tools are adequately tailored for this population). Only a small minority of interventions made use of peer specialists; but in these studies, the peer component was evaluated as very promising (e.g., [64,77]), suggesting a strong potential in this area of patient support.

Two thirds of the studies designed to optimize the health care utilization of individuals with a SMI had a duration of 6–12 months, which is a relatively short time period, in perspective of the special need of this population for enduring interventions [41]. Some studies considered follow-up measurements, but only three of them, after the end of intervention. However, this is an important factor, since changes in physical health often improve gradually and behavioral changes tend to vanish [106,107]. In terms of the intervention frequency, we found a significant number of studies reporting no information at all. This lack of provided information is suboptimal since systematic and empirical overviews need detailed information to generate meaningful data.

Concerning the research question of promising effects by types of interventions and concepts, it is not easy to relate the included interventions differentially to positive outcomes on physical health and utilization behavior. Nearly every identified study involved a complex health intervention, which makes it difficult to disentangle specific study elements and their contribution to the effects in the outcome parameters. Nearly all kinds of interventions reported at least some positive outcomes, and further quantitative analyses were not subject of the present scoping review. However, some studies we found particularly promising, as they used robust research designs [54,56,57,77]. The main interventions in these studies comprised individual care-management support, additional group settings, training for the better use of health care using motivational and self-management techniques, as well as the establishment of a health screening feedback loop. We hypothesize, in general, that interventions with longer durations, a higher frequency, and the involvement of multifactorial elements on the health care system-, provider-, and patient-levels, tend to be more promising (see [87]); but this has to be tested in further (quantitative) studies.

Remarkably, only ten out of 33 eligible studies reported an improvement in the health service utilization itself (i.e., the main intervention target). Thus, in future designs and analyses, the observation of service use per se (at best not only via self-report but also using administrative routine data) would be a desirable outcome measure.

Evidence-based interventions in health should be designed using a well-established psychological, behavioral, and implementation framework that is used to guide the intervention [6,50,51,108]. We included this aspect in our scoping review and found the explicit use of a specific framework in about 75% of the studies (based on formerly evaluated interventions, adapted complex care models, such as the chronic care model or the assertive community treatment model, or different behavioral, motivational, social, and cognitive theories). This is promising, compared to findings of other reviews in the field of integrated care, but still supports the recommendation to use (comprehensive) care models to inform the development and evaluation of future interventions for people with a SMI [51]. Further potential for improvement could be a more explicit focus on shared decision making and shared responsibility, as well as on the empowerment in the field of physical health care [109]. As interventions to promote physical health in people with a SMI tend to be of a complex nature, the need for a structured and evidence-based implementation and participatory research process seems crucial [108]. Interestingly, only a small number of studies included in this review reports the use of such a framework, even if evidence-based frameworks are available [110,111,112].

### 4.3. Research Gaps and Future Research Directions

We identified a number of research gaps and derive recommendations for future action in the area of promoting physical health care utilization among people with a SMI, beginning with a plea for studying this topic at all. It can be assumed that more than the identified health behavior promotion studies exist that are not subject to accompanying research and evaluation. The conceptualization of the identified studies suggests rather a pilot character of these interventions than a permanently implemented structure in the respective health care system (although we cannot know whether some of the interventions of our study literature have, in the meantime, been integrated in regular health care). Not only RCTs but also naturalistic observational studies are needed, including the use of administrative routine data with larger naturalistic samples and sufficient statistical power to detect population effects, as well as long-term follow-up (especially with regard to excess mortality) [46].

If studied, interventions (and their underlying rationales) should be described more precisely, in order to investigate the mediating variables of the effects. Further limitations of the present literature refer to the problem of the “common method bias” (such as the dependence of intervention and source of outcome measurement; Podsakoff et al., 2003). Here, especially the objective measurement (not self-report only) is needed, proximal in terms of the utilization enhancement, as well as distal, in terms of physical health risk factors and outcomes.

Concerning the target group, important patient populations have been hardly addressed, namely older SMI patients, people with borderline personality disorder, or people with (comorbid) substance use disorders, who have differential needs within the spectrum of SMI [6,32,40,46,47,113].

Finally, the community based participatory research should be established to increase the likelihood that treatment preferences and special issues associated with the stigmatization of ill mental health and sociocultural barriers are addressed adequately [114].

### 4.4. Strengths and Limitations

The limitations include the restriction to articles published in the English and German languages which might have contributed to the under-representation of studies conducted outside of the U.S. and Europe. The generalizability of the presented interventions is also limited, due to the very different health care systems and resources. Thus, much more context-related research is needed to cover the full picture of our topic. Further, we did not provide a systematic evaluation of the study quality and the quantitative aggregation of effectiveness of the identified interventions; but this would have been beyond the scoping nature of this review [42].

The present scoping review followed the standards for scoping reviews including the preregistration and presentation of results. We tried to optimize the selection and discussion process as a multi-professional study team (psychology, psychiatry, health services research, health, and social science) collaborating in a project aiming at the physical health promotion of people with a SMI (PSY-KOMO [115]). We tried to map the existing literature on the promotion of the utilization of physical health care for people with a SMI in detail, and to provide comprehensive supplemental material, in order to identify research gaps and to support subsequent studies.

## 5. Conclusions

Promoting access to and the utilization of primary and specialized physical health care for people with severe mental disorders, is a pragmatic and cost-effective measure to enhance the health equity in this vulnerable and often neglected patient population. However, such interventions still seem to be rare, or at least not complemented by evaluation studies documented in the literature. Promising efforts in this important section within the field of integrated care exist, mostly based on theoretical models. Most of them seem to be pilot in nature and are not an investigation of the sustained large-scale implementation in their respective health care system.

The present review provides a map of evidence, summarizing the respective studies along with the applied interventions, rationales (if reported), and outcomes. This can serve as a starting point for further quantitative evaluation of the effectiveness of this kind of behavioral intervention. We call for expanding efforts to study the issue of the physical health of individuals with a SMI in routine health care settings and shifting the focus to promoting the referral of people with a SMI to—and the use of—already existing prevention and health care resources for the general population.

## Figures and Tables

**Figure 1 ijerph-20-00126-f001:**
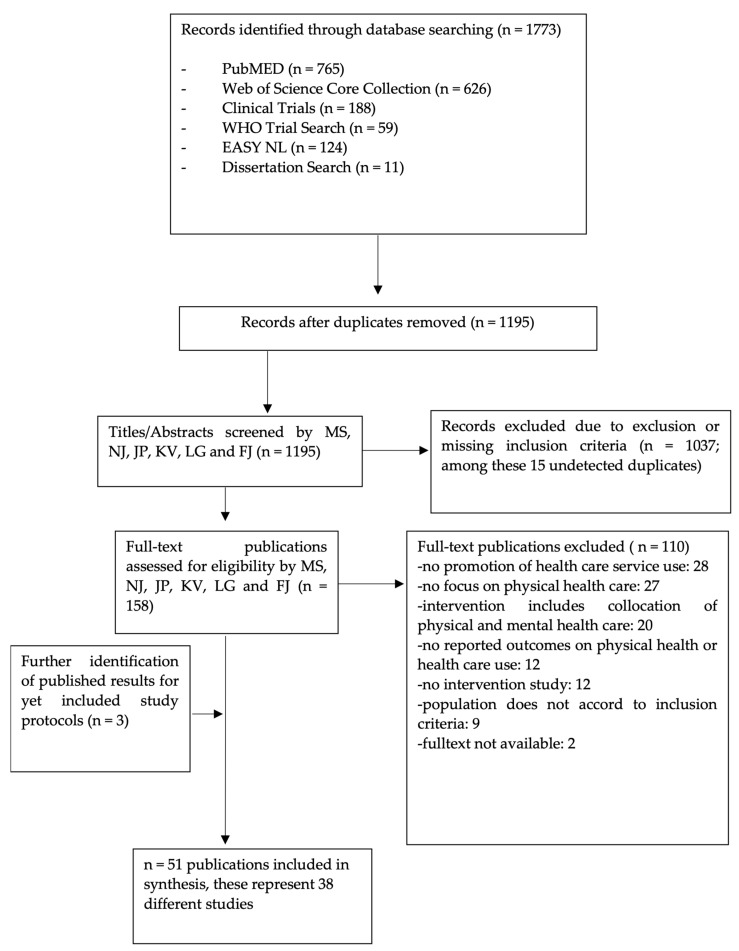
Study selection and the exclusion process (PRISMA flow diagram).

**Table 1 ijerph-20-00126-t001:** Types and content of the interventions to promote the utilization of physical health services for people with a SMI. (Note: Additional interventions not predominantly targeting the utilization of physical health services are written in italics; listed types and content of interventions are not mutually exclusive).

Type/Content of the Study Intervention	Studies (Study Identification Number from Appendix A, See Appendix A)	Number of Studies (n = 38)
Delivery involves individual setting	1, 2, 3, 5, 6, 7, 8, 9, 10, 11, 12, 13, 14, 15, 16, 17, 18, 19, 20, 21, 22, 23, 24, 25, 26, 27, 28, 29, 30, 31, 32, 33, 34, 35, 36, 37	37
Delivery involves group setting	1, 4, 5, 6, 7, 13, 15, 17, 18, 19, 25, 26, 27, 31, 32, 36	17
Phone maintenance	1, 29	2
App-based intervention	34	1
Peer-led intervention	15, 17, 22, 24	4
Peer-specialist involvement	4, 13	2
Care-management	2, 3, 5, 6, 8, 9, 11, 12, 13, 14, 16, 18, 20, 21, 22, 23, 24, 25, 26, 27, 28, 31, 32, 33, 35, 36, 37	27
Self-management training	1, 4, 6, 7, 10, 11, 12, 13, 14, 15, 16, 17, 18, 20, 22, 24, 25, 26, 27, 28, 29, 30, 34	23
Physical health education	4, 5, 11, 13, 14, 15, 17, 18, 19, 23, 25, 26, 27, 28, 31, 33	16
Health screening and monitoring	5, 6, 7, 10, 11, 12, 20, 21, 29, 30, 31, 32, 34, 35, 36, 37	16
Distinct training to improve health care utilization	1, 2, 4, 5, 7, 8, 13, 15, 16, 17, 23, 24, 30	13
Care-plan development	7, 22, 25, 26, 27, 29, 30, 35, 37	9
Motivational support	2, 7, 12, 14, 16, 19, 30	7
Treatment adherence support	2, 7, 12, 15, 17, 20	6
Problem solving training	1, 7, 14, 22, 37	5
Improvement of health careinformation interface	3, 11, 33, 38	4
Empowerment	3	1
Stigma-reduction	27	1
Structured doctor and nurse visits	3	1
*Lifestyle changes*	1, 12, 15, 17, 19, 23, 29, 30, 31, 32, 34, 36	12
*Mental health education*	1, 5, 6, 9, 17, 18, 25, 26, 27, 37	10
*Staff training*	3, 4, 18, 25, 26, 27, 33, 35, 38	9
*Wellness enhancement*	5, 13, 28, 31	4
*Involvement of social network*	8, 28, 37	3
*Crisis intervention*	8	1
*Critical appraisal of medication*	36	1
*Local implementation customization of intervention*	38	1
*Support in daily living*	3	1

**Table 2 ijerph-20-00126-t002:** Reported outcome measures in the studies including interventions to promote the utilization of physical health care for people with a SMI.

Outcome Measure	Studies (Study Identification Number from Appendix A, See Appendix A)	Number of Studies (n = 38)
Self-report based data	1, 2, 3, 4, 5, 6, 7, 9, 10, 11, 12, 13, 14, 15, 16, 17, 18, 19, 20, 22, 23, 24, 25, 26, 27, 28, 29, 30, 31, 32, 33, 34, 35, 36, 37	35
Thereof: no other data-sources than self-report based data	10, 13, 15, 17, 24, 25, 27, 31, 37	9
Physiological measures	1, 7, 9, 12, 14, 18, 19, 20, 21, 22, 23, 26, 29, 30, 32, 34, 35, 36	18
Administrative data	3, 8, 11, 16, 21, 28, 29, 33, 35, 38	10
Participation data (e.g., attendance rate)	6, 11, 22, 33, 34	5
Behavioral assessment	2, 4	2

**Table 3 ijerph-20-00126-t003:** Study authors’ interpretation of the success of the interventions to promote the utilization of physical health care for people with a SMI.

Study Authors’ Interpretation of the Intervention Success	Studies (Study Identification Number from Appendix A, See Appendix A)	Number of Studies (n = 33)
Improvement of other physical health-behavior related outcomes, besides service utilization	1, 2, 4, 6, 11, 13, 14, 15, 17, 24, 30, 32, 34, 37	14
Improvement in physical health outcomes	2, 5, 9, 12, 14, 15, 18, 25, 26, 30, 32, 36, 37	13
Improvement in health service utilization	3, 5, 6, 9, 11, 17, 21, 24, 33, 38	10
Improvement in mental health outcomes	1, 2, 3, 5, 15, 16, 18, 32, 36, 37	10
No positive physical health related outcomes at all	10, 19, 23, 27, 28, 31	6
Reduction of emergency department use	5, 6, 8, 24	4

**Table 4 ijerph-20-00126-t004:** Theoretical rationale or model of interventions to promote the utilization of physical health care for people with a SMI.

Theoretical Rationale or Model	Studies (Study Identification Number from Appendix A, See Appendix A)	Number of Studies (n = 38)
Based on evidence-based intervention (s)	1, 2, 4, 5, 6, 10, 11, 12, 13, 14, 15, 16, 17, 19, 30, 35, 37	17
Adaption of a care model or framework of healthcare	3, 5, 7, 9, 12, 18, 23, 24, 25, 26, 27, 29, 33	13
Based on established theory	1, 2, 14, 18, 20, 21, 23	7
Use of a study implementation framework	10, 22, 27, 29, 35, 38	6
No distinct model or theoretical rationale	8, 22, 28, 31, 36	5
Based on eclectic empirical evidence	21, 32, 34, 38	4

## Data Availability

Data sharing is not applicable [comprehensive presentation of the review process and content in the present article and its Appendix A].

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
