# Peer review of "Interventions to Promote the Utilization of Physical Health Care for People with Severe Mental Illness: A Scoping Review"

_ijerph, 2022, doi:10.3390/ijerph20010126_

Round 1
Reviewer 1 Report
Overall:
The scope of promotion for service use as a sub-concept of integrated care is not well investigated. Therefore, The manuscript concerns a topic relevant to clinical practice. The introduction clearly describes the gap in the current research. The methodology clearly describes the inclusion and exclusion criteria and the steps taken in the research process.
Overall I do have some (minor) comments:
Tables ables offer an overview but little insight in actual interventions.
The discussion section could provide some more in depth insights. For example: As reported, some studies showed an improvement in health service utilization, but neither the results or discussion section clearly states which types of interventions created these results. Other topics, such as implementation strategies, care management, staff training, … could also benefit from a more descriptive approach on what exactly is meant by these concept.
L52: no reference number 9?
L50: Interventions which mainly focus on the patient side to directly improve physical health of people with SMI have often been explored in terms of life-style interventions. Please elaborate on these lifestyle interventions, as they are mostly limited to educational interventions, and effectivity remains unclear. In other words, underline the relevance for mentioning this.
L55: The term equal is used, equitable or equity in general seems to be more relevant. Please do not mix up the terms equity and equality, as they differ profoundly in meaning with regard to the scope of the article.
L57: A reference seems to be lacking
L67: The PCC approach is well described in the methodology section, to avoid repetition and/or misinterpretation, it seems more logical to elaborate on the PCC in the methodology section. For example, the introduction mentions ‘health promotion programs’ as an inclusion criterium for concepts, but this is not mentioned in the methodology.
L102: Why are interventions on a provider level excluded? Couldn’t this possibly increase the promotion of access by health care workers? The reason of exclusion seems to be missing.
L109: Present tense is suddenly used.
L136: For me, it is unclear what has been done by ‘contacting experts in the field’. Isn’t this beyond the scope of a review?
L139: Why only the first 100 articles?
L209: The methodology section states that (educational) interventions targeting health staff were excluded, yet staff training is included in the list of interventions.
L239: This sentence is phrased in an unclear manner
L247: Why is there an emphasis on mental health outcomes in the results section, as it was stated in the exclusion criteria (L102)
Reference list: Avoid double references: L419 & L427; L422 & 439; L453 & 466
Author Response
Manuscript ijerph-2094987 [Interventions to promote utilization of physical health care for people with severe mental illness: A scoping review]
Reply to Reviewer #1
Dear Reviewer,
We thank you for your comments that help to improve the manuscript in several ways! Please find our point-by-point reply below (and actual changes made in the manuscript can be found in the attached files ijherph_manuscript_v2_track-changes and ijherph_manuscript_v2 with the adopted changes):
The scope of promotion for service use as a sub-concept of integrated care is not well investigated. Therefore, the manuscript concerns a topic relevant to clinical practice. The introduction clearly describes the gap in the current research. The methodology clearly describes the inclusion and exclusion criteria and the steps taken in the research process.
Overall, I do have some (minor) comments:
- Tables offer an overview but little insight in actual interventions.
As the tables included in the manuscript itself are presented in a more aggregated way (space restrictions), we added a specification in the respective results section detailing that insights in actual interventions of a respective study can be found in the comprehensive (50 pages) supplemental table S7:
“Each identified intervention type is presented with the respective studies in Table 1. As the implementation of the intervention types differs in each study, details of a particular study can be found in Supplementary Table S7. For example, the intervention type ‘training on better utilization of health care’ in study 1 was part of nine group training sessions delivered by health care professionals and peer specialists, targeting mainly the ability to communicate with health care providers. In study 23, this type of intervention was implemented through weekly and individual coaching sessions delivered by health professionals. These helped participants improve their ability to engage with healthcare providers to address their health comorbidities.”
- The discussion section could provide some more in-depth insights. For example: As reported, some studies showed an improvement in health service utilization, but neither the results nor discussion section clearly states which types of interventions created these results. Other topics, such as implementation strategies, care management, staff training, … could also benefit from a more descriptive approach on what exactly is meant by these concepts.
Thank you for this remark – in fact this question on which effect can be attributed to what intervention is also very important to us. However, given the studies we have, this question cannot be answered properly (i.e., no dismantling designs etc. exist in that field). Nearly every study identified included a complex health measure, making it difficult to attribute specific study elements to specific outcome parameters. However, without stating any evidence, we added elements of the (in our regard) most promising studies in the discussion section and added a sentence addressing the problem (“Nearly every identified study involved a complex health intervention, which makes it difficult to disentangle specific study elements and their contribution to effects in the outcome parameters.“)
Regarding your second notice about a more descriptive approach to distinct topics, we want to refer to the comprehensive supplemental table S7, as the realization of a topic differs from study to study. For example, the implementation strategy in study 22 was based on a community-based participatory research approach.
- L52: no reference number 9?
Thank you for this notice, we fixed this problem regarding our Zotero formatting.
- L50: Interventions which mainly focus on the patient side to directly improve physical health of people with SMI have often been explored in terms of life-style interventions. Please elaborate on these lifestyle interventions, as they are mostly limited to educational interventions, and effectivity remains unclear. In other words, underline the relevance for mentioning this.
The concept of lifestyle interventions includes structured approaches to change physical activity, weight, healthy diet, and healthy habits (e.g., reduce or quit smoking), which we outlined now in L52 (“These interventions focus on promotion of healthy lifestyles and wellness, including structured approaches to change physical activity, weight, healthy diet, and healthy habits (e.g., quit or reduce smoking).”). We mentioned this topic (among others) to give an overview about already existing reviews in the field, as suggested by the guidelines for scoping reviews.
- L55: The term equal is used, equitable or equity in general seems to be more relevant. Please do not mix up the terms equity and equality, as they differ profoundly in meaning with regard to the scope of the article.
Thank you for this correction, we replaced equal with equitable.
L57: A reference seems to be lacking
Thank you for this notice, we added the according reference.
- L67: The PCC approach is well described in the methodology section, to avoid repetition and/or misinterpretation, it seems more logical to elaborate on the PCC in the methodology section. For example, the introduction mentions ‘health promotion programs’ as an inclusion criterium for concepts, but this is not mentioned in the methodology.
Thank you for this notice. We added ‘health promotion programs’ in the methods section and shortened the introduction regarding the PCC approach.
- L102: Why are interventions on a provider level excluded? Couldn’t this possibly increase the promotion of access by health care workers? The reason of exclusion seems to be missing.
Thank you, that indeed needs to be clarified. We wanted to focus on patient-centered interventions to ensure basic comparability of studies. Since the concept under study focuses on the utilization behavior of individuals, studies with an intervention solely at the provider level (without involving the target population) would not fit well with this focus. Interventions that involved health care workers were not excluded if the intervention program also included patient-centered interventions.
We therefore specified study aim 1: „How many and what types of patient-centered interventions have been examined or designed to promote utilization of physical health care systems for people with SMI?” and added the reason of exclusion for studies which only address physical health of people with SMI on the provider-level (L107).
- L109: Present tense is suddenly used.
Thank you for this notice, we changed this into past tense.
- L136: For me, it is unclear what has been done by ‘contacting experts in the field’. Isn’t this beyond the scope of a review?
The contact with experts was not structured and was only an additional aid in our process. It had no (major) influence on the scoping review, and the sentence can therefore be deleted without any problems.
- L139: Why only the first 100 articles?
We discussed the first 100 articles in the entire author group to optimize the evaluation of inclusion and exclusion criteria and consensus building. Thereafter, all articles were reviewed by only two reviewers.
In case the respective pair of reviewers had a disagreement, consensus was ensured for all articles (including the first 100) by WH, who acted as a third reviewer.
We added the notion in the manuscript that the first 100 article review was a “training phase” (L147).
- L209: The methodology section states that (educational) interventions targeting health staff were excluded, yet staff training is included in the list of interventions.
Thank you for the notice – also this needs clarification in order to avoid irritation and this also fits to point 8. above.
Studies usually investigated complex interventions, which means that promotion of health care utilization was not necessarily the only intervention element. The concept of promotion of health care utilization had to be part of every study we included. Studies were excluded if their interventions were restricted to the provider level (e.g., exclusive staff training, see above in 8.). Additional interventional elements like staff training were not an exclusion criterion.
We specified this in the methods (L102) and results section (3.3) and also adapted Table 1 to make this clearer.
- L239: This sentence is phrased in an unclear manner
Thank you for this notice. We specified the sentence („We identified 18 studies who included physiological measures such as cardiovascular health parameters.”)
- L247: Why is there an emphasis on mental health outcomes in the results section, as it was stated in the exclusion criteria (L102)
This is consistent with items 8. and 12. above. Because the identified studies generally examined complex interventions, we also reported outcomes that went beyond the actual promotion of health service use. Given the nature of a scoping review, we wanted to be as comprehensive as possible about the included studies, which included reporting all findings (even if they did not relate to our main topic). To avoid further irritation, we added, "To include mention of effects beyond health service utilization ...".
- Reference list: Avoid double references: L419 & L427; L422 & 439; L453 & 466
Thank you for this very attentive and important note. We fixed the doubled references of L419 & L427; L422 & 439; L453 and double-checked for further duplicates. L453 and L456 (did you mean L456 instead of L466) are two different studies (10.1016/j.jagp.2013.04.013. and 10.1176/appi.ps.201300023.)

Reviewer 2 Report
Thank you for the opportunity to review this very well-written and robust review. This work includes a significant contribution to the literature. A timely review for studies that focused on interventions to promote utilization of physical health care for people with severe mental illness was conducted. I have one particular issue that needs to be addressed to further clarify the findings.
The categorization that authors used in their findings does not seem to be mutually exclusive and thus, is a little confusing. For example, in Table 1 including a list of types of interventions to promote utilization of physical health care for people with SMI, a 'health education' category can also be included within either the 'individual support' or 'group intervention' depending on the relevant study's scope right?
Also, how is 'psychoeducation' different from 'health education'?
Authors should not use the raw description of the intervention as in the original papers. Rather, a thorough thematic analysis should be conducted to better classify retrieved interventions based on their content.
Also in Table 1, the list seems premature. Authors list the intervention type for some studies (ex., case management, education, structured visits) while they mention the major outcomes in others (ex., Wellness, problem solving).
Same comment goes for table 3. What do you mean by “Thereof only self-reported data” ? Also , “follow up measurement” can be any of the other categories.
In table4, what do you mean by “health-behavior related outcomes” ? this should be clarified as either physical, mental or both, to be in line with other categories.
Finally, I strongly recommend authors to present their findings in same order as in their research questions. So that the results section will show the answer for each of the study’s question clearly.
Thank you
Author Response
Manuscript ijerph-2094987 [Interventions to promote utilization of physical health care for people with severe mental illness: A scoping review]
Reply to Reviewer #2
Dear Reviewer,
We thank you for your comments that help to improve the manuscript in several ways! Please find our point-by-point reply below (and actual changes made in the manuscript can be found in the attached files ijherph_manuscript_v2_track-changes and ijherph_manuscript_v2 with the adopted changes):
Thank you for the opportunity to review this very well-written and robust review. This work includes a significant contribution to the literature. A timely review for studies that focused on interventions to promote utilization of physical health care for people with severe mental illness was conducted. I have one particular issue that needs to be addressed to further clarify the findings.
- The categorization that authors used in their findings does not seem to be mutually exclusive and thus, is a little confusing. For example, in Table 1 including a list of types of interventions to promote utilization of physical health care for people with SMI, a 'health education' category can also be included within either the 'individual support' or 'group intervention' depending on the relevant study's scope right?
Thank you for this remark – this indeed has to be clarified in order not to confuse further readers. We made the following revisions in this regard:
- Reorganization of table 1 (description, structure): differentiation of delivery setting, interventions in the context of promotion of health care service use, interventions with other focus; we further added a note that intervention categories are not mutually exclusive
- Regarding the nature of complex health interventions, we mainly identified studies which did not only promote service use. As we wanted to provide a comprehensive overview about the included studies, other interventions are listed as well. To clarify this we added some explications in the results section describing table 1.
- Also, how is 'psychoeducation' different from 'health education'?
Thank you for the notice. We changed the description of both terms into “physical health education” and “mental health education” in table 1 as well as results section in order to prevent confusion.
- Authors should not use the raw description of the intervention as in the original papers. Rather, a thorough thematic analysis should be conducted to better classify retrieved interventions based on their content.
This is a rather difficult topic that we cannot really address, as we do not believe that further specific thematic analysis (e.g., using the six-step method) is possible (but hopefully not necessary). We proceeded as follows: In our comprehensive (50 pages) Supplementary Table 7, we compiled all study information into the categories (raw descriptions in this table were used only when appropriate and necessary). This time-consuming process was followed by a recheck by one of the other reviewers to ensure that all information was recorded correctly. We then built the aggregated tables 1 - 4 on the information in this comprehensive table. To do this, we listed each intervention type of each study in comprehensive Excel spreadsheets, removed synonymous intervention types in a second step, aggregated the remaining intervention types in a third step, and reapplied the resulting list of intervention types to each study in a final step (corresponding Excel spreadsheets are available upon request).
We feel that it is not really helpful to include this detailed information in the methods section under "Study Synthesis", but if you consider this mandatory, we would of course add it.
- Also in Table 1, the list seems premature. Authors list the intervention type for some studies (ex., case management, education, structured visits) while they mention the major outcomes in others (ex., Wellness, problem solving).
Thank you for this note which confirms the necessity of the revision of table 1 we made according to your request in point 1. As described in the answer to your first note, we specified the descriptions. “Wellness” and “problem solving” can easily be misinterpreted as outcomes, so we changed these for example into “wellness enhancement” and “problem solving training”.
- Same comment goes for table 3. What do you mean by “Thereof only self-reported data” ? Also , “follow up measurement” can be any of the other categories.
We also thank you for these comments, which help to avoid confusion. We specified table 3, changing “Thereof only self-reported data” into “Thereof: no other data-sources than self-report based data”.
In order to prevent confusion, we removed “follow up measurement” as a category from the table and described it in the text, as this category describes the data assessment design and not the type of outcome. This was also an opportunity to correct an error (three studies used follow-up measurement, not two).
- In table4, what do you mean by “health-behavior related outcomes” ? this should be clarified as either physical, mental or both, to be in line with other categories.
Thank you – we added “physical” respectively.
- Finally, I strongly recommend authors to present their findings in same order as in their research questions. So that the results section will show the answer for each of the study’s question clearly.
Thank you for this valuable comment we addressed as follows:
- To ensure that the results are in the same order as our research questions, we changed Table 2 and the corresponding section to Table 4, Table 3 to Table 2, and Table 4 to Table 3.
- In addition to our original research questions, we have determined that it is useful to include theoretical rationales of the relevant studies in our review (L160); this is the reason we include the table "Theoretical rationale or model of interventions..." at all.
- The answer to our third research question arises from the discussion (and thus naturally comes last in the presentation)
